



# Migrating tide climatologies measured by a high-latitude array of SuperDARN HF-radars

Willem E. van Caspel[1,2], Patrick J. Espy[1,2], Robert E. Hibbins[1,2], and John P. McCormack[3]

[1]Department of Physics, Norwegian University of Science and Technology (NTNU), Trondheim, Norway,
[2]Birkeland Centre for Space Science, Bergen, Norway,
[3]Space Science Division, Naval Research Laboratory, Washington DC. USA

**Correspondence:** W.E. van Caspel (willem.e.v.caspel@ntnu.no)

**Abstract.** This study uses hourly meteor wind measurements from a longitudinal array of 10 high-latitude SuperDARN HF-radars to isolate the migrating diurnal, semidiurnal and terdiurnal tidal modes at Mesosphere-Lower-Thermosphere (MLT) heights. The planetary-scale array of radars covers 180 degrees of longitude, with eight out of 10 radars being in near-continuous operation since the year 2000. Time series spanning 16 years of tidal amplitudes and phases in both zonal and meridional wind are presented, along with their respective annual climatologies. The method to isolate the migrating tidal modes from SuperDARN meteor winds is validated using two years of winds from NAVGEM-HA (Navy Global Environmental Model - High Altitude). The validation steps demonstrate that, given the geographical spread of the radar stations, the derived tidal modes are most closely representative of the migrating tides at 60°N. Some of the main characteristics of the observed migrating tides are that the semidiurnal tide shows sharp phase jumps around the equinoxes and peak amplitudes during late summer, and that the terdiurnal tide shows a pronounced secondary amplitude peak around DOY 260. In addition, the diurnal tide is found to show a bi-modal circular polarization phase relation between summer and winter.

## 1 Introduction

Atmospheric tides are global scale waves excited primarily by radiative and latent heating effects in the troposphere and stratosphere (Chapman and Lindzen, 2012). The tides have a latitudinal Hough-mode and longitudinal zonal wavenumber (S) structure, and in the absence of dissipation their amplitudes increase exponentially with altitude due to the decreasing density of the atmosphere. In the mid- to high-latitude Mesosphere-Lower-Thermosphere (MLT), tides are an important driver of short- and long-term variability in the winds, temperatures, and densities (Smith, 2012). The migrating diurnal (DW1; for Diurnal, Westward, S=1), semidiurnal (SW2), and terdiurnal (TW3) tides are most closely tied the daily insolation cycle, following the apparent motion of the sun with a period of oscillation of 24, 12, and 8 hours, respectively. Non-migrating tides are waves whose period of oscillation are also an integer fraction of a solar day, but whose phase velocities are not sun-synchronous.

Observations capable of separating the longitudinal structure of the migrating tides from the non-migrating components have remained sparse, with the exception of satellites (e.g., Garcia et al., 2005; Ortland, 2017; Pancheva and Mukhtarov, 2011). Typical drawbacks associated with satellite measurements arise due to yaw-cycle intermittency, as well as from constraints imposed by asynoptic sampling (Salby, 1982). Single station tidal measurements using MF, HF, or VHF-radars have been





numerous (Reid, 2015), but because they lack longitudinal coverage, the migrating and non-migrating tides are aliased to a single local wave with integer fraction of a solar day period. Such spatial aliasing is especially problematic when migrating and non-migrating tides are known to have different seasonal cycles (Sakazaki et al., 2018; Hibbins et al., 2019). A planetary-scale longitudinal chain of time-synchronized measurements can potentially bypass most, if not all, of the drawbacks associated with satellite and single station tide measurements, albeit along a single latitude. The array of SuperDARN (SD) radars used in this

study is unique in that it covers 180 degrees of longitude along a latitude band centered around 60°N, and that eight of the 10 radars have been providing hourly meteor wind measurements of the MLT near-continuously since the year 2000. As a result of the simultaneous temporal and spatial sampling by the SD radars, unambiguous amplitudes and phases of the migrating diurnal, semidiurnal and terdiurnal tides can be isolated.

The following section gives a description of the data and method used to extract the migrating tides from the SD meteor

winds. Section 3 presents time series spanning 16 years of hourly tidal amplitudes and phases in both the zonal and meridional winds, in addition to their respective annual climatologies. The method to extract migrating tidal modes from the SD meteor winds is validated in section 4 by means of sampling experiments with winds from NAVGEM-HA (Navy Global Environmental Model - High Altitude), addressing the geographical spread and changing availability in time of the SD radars. Lastly, the results are discussed in section 5.

## 40  2   Data and methodology

### 2.1   SuperDARN meteor winds

Figure 1 shows the geographical location and data availability between the years 2000 and 2016 of the 10 SD radars used in this study. The SD radars operate in a 10-15 MHz frequency band and are designed to measure ionospheric E- and F-region plasma phenomena. However, they also detect near-range meteor echoes in the first four range gates that can be used to determine

neutral horizontal wind velocities (Hall et al., 1997). The phase shift of the return signal of each meteor echo is a measure of the component of the neutral wind velocity along the line of sight. An hourly mean horizontal wind vector is constructed from the aggregate line of sight wind vectors, over a 45 degree spread in azimuth, using a Singular Value Decomposition (SVD). While the line of sight velocities are typically very well defined (errors below 1 ms$^{-1}$, Chisham and Freeman, 2013), the SVD is applied only to line of sight velocities having a signal to noise ratio greater than 3.0 dB and spectral width of at most 25

50  ms$^{-1}$, to reduce contamination by sources such as auroral and sporadic E-region echoes. In addition to a hourly horizontal wind vector, the SVD also yields the standard deviation of the hourly winds, which typically ranges between 5-15 ms$^{-1}$ for the meridional wind and 10-30 ms$^{-1}$ for the zonal wind. The seasonal mean vertical distribution of meteor echoes as observed by the SD radars is a Gaussian centered on 102-103 km altitude, extending from approximately 75 to 125 km altitude with a full width at half maximum of 25-35 km (Chisham and Freeman, 2013; Chisham, 2018). The SD meteor winds therefore represent

a broad vertical average, which in earlier studies has been found to best correlate with neutral winds measured by traditional MF and meteor radars around 95 km altitude (Hall et al., 1997; Arnold et al., 2003).



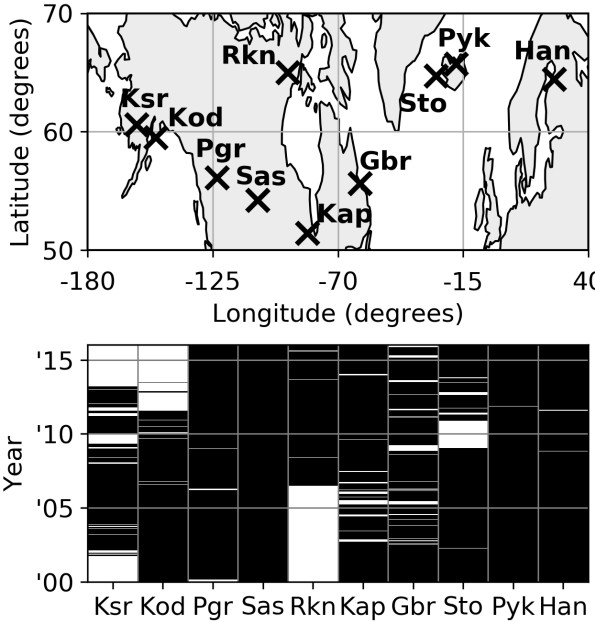

**Figure 1.** Abbreviated names and geographic locations (upper panel), and time of operation between the years 2000 and 2016 (black marking lower panel) of the SuperDARN radars used in this study.

On average each hourly SD meteor wind measurement is based on ∼700 meteor echoes. Before extracting the migrating tides, however, measurements based on fewer than 75 meteor echoes are discarded, as are those resulting from non-standard modes of operation. The latter amounts to discarding winds having absolute values above 100 ms$^{-1}$ and winds fitted with a zero standard deviation (following Hibbins and Jarvis, 2008). The lower limit on the number of meteor echoes is to ensure quality of the fitted winds. Caution has been taken to ensure that no spurious tidal signals are introduced by the quality check, which may arise due to the diurnal cycle in meteor detections. This was verified by replacing all remaining data points with a value of 1 ms$^{-1}$ and then performing spectral analysis as outlined in the following section, to confirm that tidal spectral contamination remains negligibly small.

## 2.2 Fourier analysis

The amplitude and phase of DW1, SW2, and TW3 are calculated by least-squares fitting the function $G(\lambda, t)$ in both space and time, where $G(\lambda, t)$ represents the migrating tidal modes along with a mean wind, given by

$$G(\lambda, t) = \sum_{k=1}^{3} A_k \sin(k[\Omega t - \lambda] + \phi_k) + G_0, \tag{1}$$

where $k = 1, 2, 3$ represent DW1, SW2 and TW3, respectively, $\Omega = 2\pi/24 \text{ hr}^{-1}$; $\lambda$ is the geographic longitude in radians; and $G_0$ is the mean wind. The time development is determined by fitting $G(\lambda, t)$ over a 10-day window that is stepped forward



in time with hourly steps over the range of available data. A 10-day window length is chosen such that each fit contains a proportionally sufficient number of data points to reliably extract the seasonal characteristics of the tides, without overly smoothing short-term variability.

The longitudinal spread of measurements is optimized over the range of available data to prevent skewing the fit to any particular longitude sector. To that end, the radar longitude pairs of (Ksr, Kod), (Rkn, Kap), and (Sto, Pyk) are identified. If measurements are available for both stations at any given time, measurements of either one is excluded in a manner such as to optimize the equidistant longitudinal spread of measurements. After performing the quality check and optimizing the longitudinal spread, fits are rejected if fewer than 960 hourly data points are present over the 10-day period, corresponding to an average continuous uptime of at least four radar stations. As a result of requiring a minimum of 960 hourly data points in each fit to Eq. 1, the estimated uncertainties on the fitted parameters become negligibly small (on the order of $0.5$ ms$^{-1}$ for the tidal amplitudes when employing the standard deviations of the hourly winds as an estimate of the measurement errors).

### 2.3 NAVGEM-HA

NAVGEM-HA is a data assimilation and modeling system that extends from the surface to the lower thermosphere. In addition to standard operational meteorological observations in the troposphere and stratosphere, NAVGEM-HA assimilates satellite-based observations of temperature, ozone and water vapor in the stratosphere, mesosphere and lower thermosphere (McCormack et al., 2017). NAVGEM-HA output is on a $1°$ latitude and longitude grid with a temporal frequency of 3 hours, staying above the spatial and temporal Nyquist frequency of the tides studied in this work. For comparison with ground-based instruments, vertical profiles of NAVGEM-HA analyzed winds and temperatures are converted from the model vertical grid in geopotential altitude to a geometric altitude grid. To date, NAVGEM-HA winds and tides have been shown to be in good agreement with ground-based meteor radar observations (McCormack et al., 2017; Eckermann et al., 2018; Laskar et al., 2019; Stober et al., 2019) and with independent satellite-based wind observations as reported in Dhadly et al. (2018). In the present study we employ NAVGEM-HA analyzed winds at 82.5 km altitude, staying below altitudes where sponge layer effects may impact the tides, to validate the method of extracting migrating tidal signatures from the SD meteor wind data.

### 3 Results

#### 3.1 16 year time series

Figure 2 and 3 show the amplitudes and phases of the DW1, SW2, and TW3 tidal modes retrieved from SD zonal and meridional meteor winds between the years 2000 and 2016. Here the phases are shown as the local time of maximum (LTOM), and phases for tidal amplitudes less than 1.5 ms$^{-1}$ are not shown for sake of clarity.

SW2 shows a strongly repeatable seasonal cycle, where amplitudes peak around late summer and mid-winter, and where sharp phase jumps occur around spring and fall equinox. The late summer amplitude maximum of SW2 typically reaches values between 19-25 ms$^{-1}$. In contrast, the mid-winter amplitude maximum typically lies between 10-14 ms$^{-1}$. SW2 consistently




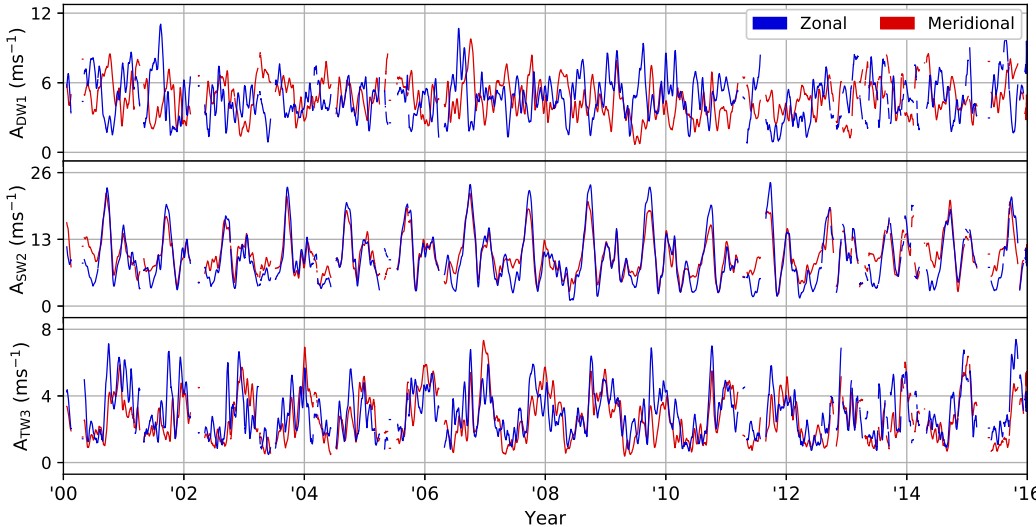

**Figure 2.** Amplitude of DW1 (top), SW2 (middle), and TW3 (bottom) in SuperDARN zonal (red) and meridional (blue) meteor winds between the years 2000 and 2016.

reaches amplitude minima coincident with the equinoctial phase jumps. While the amplitude and phase progression between the zonal and meridional components are nearly identical, meridional amplitudes can at times be 4-5 ms$^{-1}$ larger, especially during mid-summer. In terms of the absolute phase separation, the meridional component of SW2 is found to lead the zonal by

2.48 hrs on average, giving it a 32 minute offset relative to a perfect circular polarization.

TW3 also shows a strongly repeatable seasonal cycle, where a broad amplitude maximum is centered on mid-winter and where the phase begins to shift to a later time after DOY 250, stabilizes around DOY 365, and then shifts back to its pre-winter value up to DOY 90. Wintertime amplitudes typically reach values between 4-6 ms$^{-1}$, whereas the tide is nearly non-existent throughout summer. At times the amplitude of TW3 can surpass those of SW2 and DW1, in particular around fall equinox

when SW2 reaches a minimum. In addition, a pronounced secondary TW3 amplitude peak is found near DOY 260, which can be more clearly seen in the climatology presented in the next section. This peak is most pronounced in the zonal wind, where it can reach amplitudes in the range of 4-7 ms$^{-1}$. In terms of its phase, TW3 is found to be nearly circularly polarized during times when the wave has an appreciable amplitude, with the meridional component leading the zonal by 1.98 hrs on average.

DW1 shows considerably more short-term and interannual variability in its amplitude, phase, and between the zonal and

meridional components. This is reflected in the correlation coefficient of $r = -0.23$ between the time series of hourly zonal and meridional amplitudes of DW1, whereas for SW2 and TW3 this is $r = 0.90$ and $r = 0.66$, respectively. There is, however, a clear seasonal cycle present in both the amplitude and phase of DW1, which can be more clearly seen in the climatology presented in the next section.

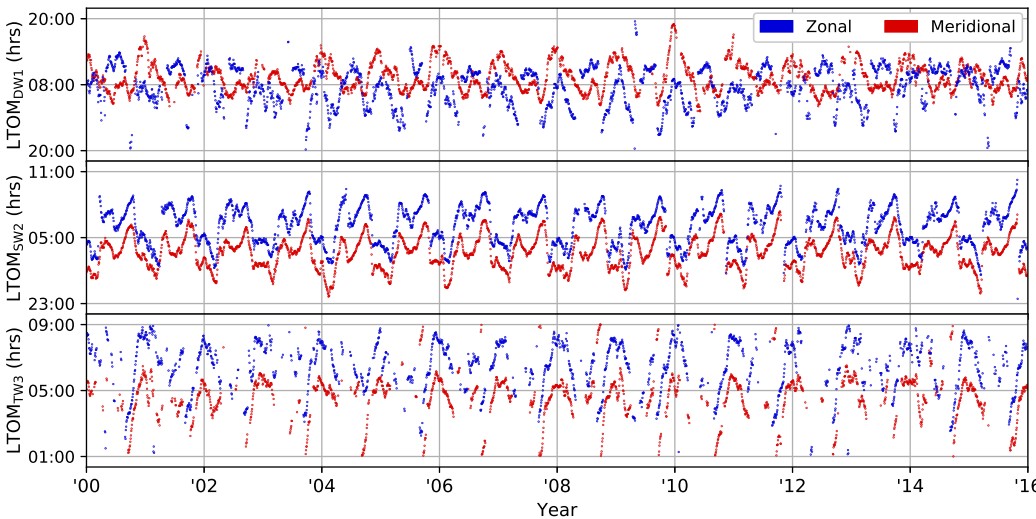

**Figure 3.** Phase of DW1 (top), SW2 (middle), and TW3 (bottom) in SuperDARN zonal (red) and meridional (blue) meteor winds between the years 2000 and 2016. Phases are plotted as the local time of maximum (LTOM). Only phases for when tidal amplitudes are greater than 1.5 ms$^{-1}$ are shown for sake of clarity.

### 3.2 Climatologies

Figure 4 shows the yearly amplitude and phase climatologies of DW1, SW2, and TW3 based on the amplitudes and phases presented in the previous section. The climatologies are constructed by calculating the mean amplitude and phase for each DOY, where the mean phase is calculated using the circular mean (Fisher, 1995). The shaded area represents the standard deviation around the climatological mean amplitude and serves as a measure of year-to-year variability.

For the zonal (meridional) component, the climatological amplitude of SW2 in late summer and mid-winter peaks at 20.7

(18.8) ms$^{-1}$ and 12.4 (12.4) ms$^{-1}$, respectively. Variability around the climatological mean of SW2 is largely constant throughout the year, with an average standard deviation of 2.2 (2.0) ms$^{-1}$. For TW3, mid-winter zonal (meridional) amplitudes peak at 4.4 (5.1) ms$^{-1}$, while the DOY 260 amplitude peaks at 5.3 (4.1) ms$^{-1}$. The average standard deviation of the amplitude of TW3 is 1.0 (0.9) ms$^{-1}$, while variability around the climatological mean is highest coincident with the amplitude peak at DOY 260 by 1.7 (1.4) ms$^{-1}$.

The climatology of DW1 stands out in that amplitudes in the meridional wind broadly tend to maximize around 6.7 ms$^{-1}$ near the equinoxes, whereas those in the zonal wind maximise around 6.7 ms$^{-1}$ near the solstices. In addition, the climatological phase shows a circular polarization where the zonal component leads the meridional by approximately 6 hrs during fall and winter, but lags it by approximately 6 hrs during spring and summer. The climatological phase thus shows a bi-modal circular polarization, with the polarization flipping sign broadly during the summer months. Variability around the climatological





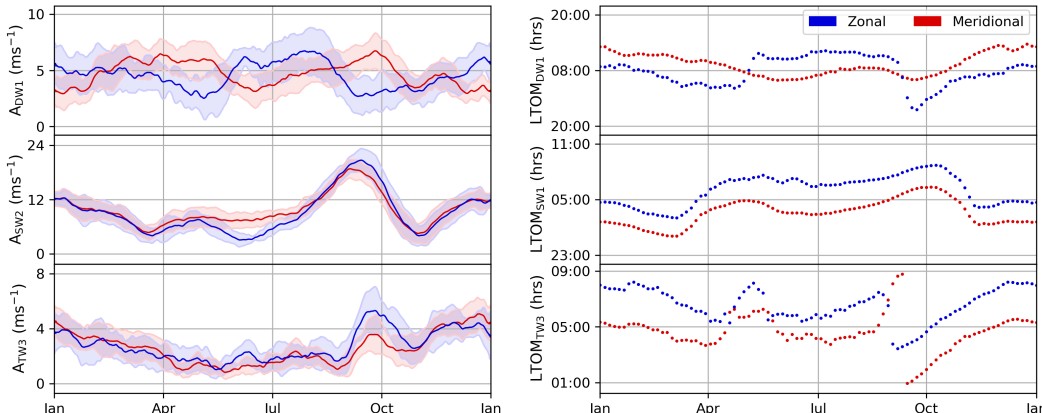

**Figure 4.** Climatologies of the amplitude (left panels) and phase (right panels) of the DW1 (top), SW2 (middle), and TW3 (bottom) based on SuperDARN meridional (red) and zonal (blue) meteor winds between the years 2000 and 2016. Shading marks the standard deviation around the climatological mean.

amplitude of the zonal (meridional) component of DW1 stays largely constant throughout the year, with an average standard deviation of 1.6 (1.3) ms$^{-1}$.

## 4   Validation

In this section sampling experiments with NAVGEM-HA are used to validate the method to extract migrating tides from the longitudinal chain of SD meteor wind measurements. The sampling experiments seek to address the geographical spread of

the SD stations, as well as the spatial sampling variations due to the changing availability of the individual stations with time (as shown in Figure 1). Cross-contamination errors arising from the geographical spread of the SD stations are expected to be large relative to the error propagating from any individual measurement uncertainties, since each fit includes at least 960 hourly data points, as discussed in section 2.2.

### 4.1   Geographical spread

To address the geographical spread of the SD stations, migrating tides (Eq. 1) are fitted to NAVGEM-HA meridional winds sampled at the locations of available SD measurements (NAVGEM-SD), after quality checking and optimizing the longitudinal spread as discussed in section 2.2. These are then compared against those fitted to a full longitude circle of data taken along 60°N (NAVGEM-360). Fits along a full longitude circle are orthogonal to any other longitudinal waves, and so form a benchmark of the 'true' migrating tides. As with the fits to SD, a 10-day time window is used where the window is now stepped

forward in 3 hourly steps to accommodate the temporal resolution of NAVGEM-HA.

    Figure 5 shows the migrating tidal modes derived from NAVGEM-SD and NAVGEM-360 for the years 2014 and 2015, demonstrating that there is no structural deviation between the two for all three tidal components. The largest amplitude de-





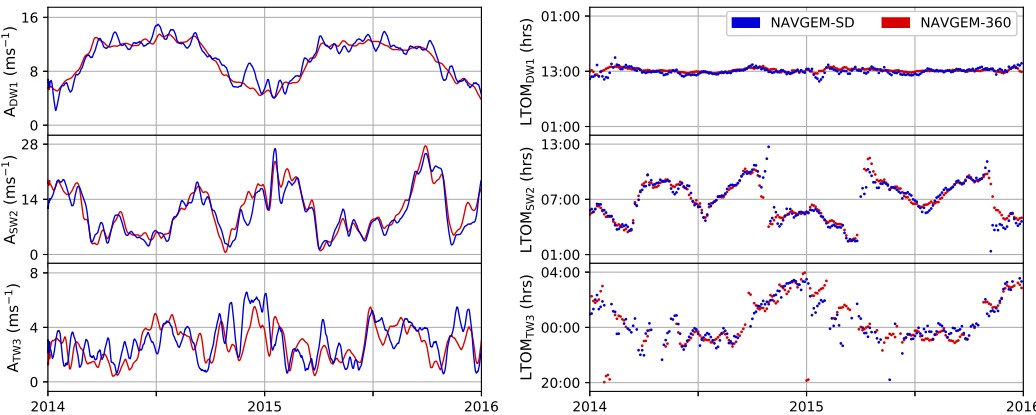

**Figure 5.** Amplitude and phase of DW1 (top), SW2 (middle) and TW3 (bottom) in NAVGEM-SD (blue) and NAVGEM-360 (red) at 82.5 km altitude for the years 2014 and 2015. Phases are plotted as local time of maximum (LTOM).

viations remain incidental, whereas the phases are in close agreement at all times. On average, the phase difference between NAVGEM-SD and NAVGEM-360 is 5.9, 5.6 and 6.0 minutes for DW1, SW2, and TW3, respectively. The geographical spread
of the SD radars is therefore concluded to not lead to significant cross-contamination errors between the tidal modes representative of the (global) migrating tides.

### 4.2 Root mean square error analysis

To examine the quality of the tides fitted to NAVGEM-SD, they are compared to those fitted to NAVGEM-360 by looking at the root-mean-square error (RMSE) between their respective tidal fields. Here the tidal fields themselves can be fully reconstructed
on a 360 degree longitude-time grid using the 3 hourly fitted amplitudes and phases. To account for the changing availability of the SD stations with time, the RMSE is reported using 2014 NAVGEM-HA meridional winds sampled at the locations of active SD stations for each year between 2000 and 2016. The resulting year-by-year RMSE values, calculated between the yearly reconstructed tidal fields of NAVGEM-SD and NAVGEM-360, are shown in Figure 6. For each year the RMSE is comparatively low relative to the absolute tidal amplitudes shown in Figure 5, assuring the validity of the method to extract the
migrating tides over the range of hourly SD data used in this study. It also shows that the stations changing with time does not induce any substantial long-term trends.

In the above, sampling NAVGEM-360 at 60°N was motivated by NAVGEM-SD most closely corresponding to NAVGEM-360 at this latitude, which is now demonstrated. To that end, the RMSE is examined between NAVGEM-SD and NAVGEM-360, where the latter is taken at each latitude between 52°N and 68°N. Figure 7 demonstrates that the RMSE for the SW2
and TW3 tidal fields reaches a clear minimum at 60°N, while the RMSE of DW1 decreases also for latitudes greater than 60°N. However, the relative difference between the RMSE of DW1 at 60°N and 68°N is comparatively low (-2.4%). The

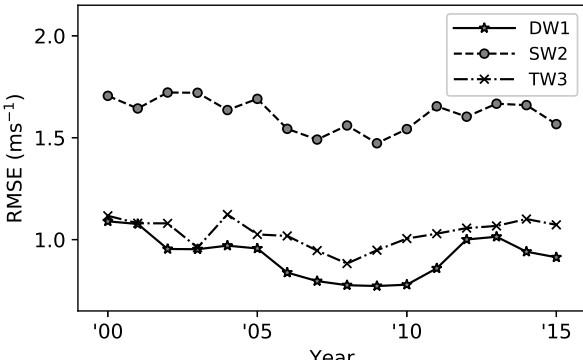

**Figure 6.** Yearly RMSE between the tidal fields constructed from fits to NAVGEM-360 and NAVGEM-SD sampled at active SuperDARN stations for each year between 2000 and 2016.

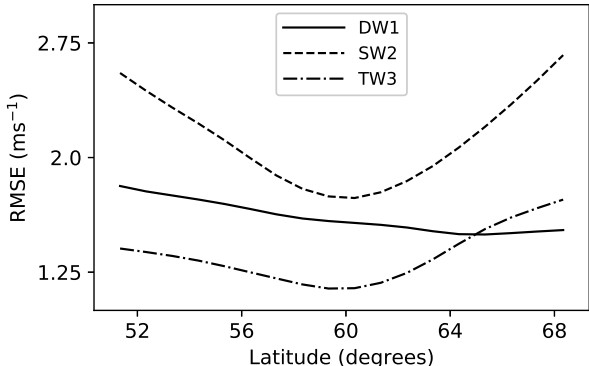

**Figure 7.** Yearly RMSE between the tidal fields constructed from fits to 2014 NAVGEM-SD and NAVGEM-360 taken along each latitude between 52°N and 68°N.

migrating tides extracted from NAVGEM-SD are therefore concluded to most closely correspond to those at 60°N. Following this conclusion, the migrating tides extracted from SD are taken to be most closely representative of those at 60°N.

## 5  Discussion

The SW2 and TW3 tidal modes isolated from 16 years of SD meteor winds show a well-defined and strongly recurring seasonal cycle. The main features of SW2, namely its amplitude peaks around late summer and mid-winter, and sharp phase jumps around the equinoxes, are in qualitative agreement with previous observational and model studies of the mid- and high-latitude migrating semidiurnal tide (Wu et al., 2011; Xu et al., 2011; Forbes and Vial, 1989). The SW2 presented in this work indicates





that the late summer amplitude peaks in the zonal and meridional winds are significantly higher than those in mid-winter, by
71% and 56% on average, respectively.

The seasonal cycle of TW3, showing a broad wintertime amplitude maximum and a near 4-hour LTOM phase progression
tracing a half-circle throughout winter, is also in qualitative agreement with previous observational and model studies of the
mid- and high-latitude migrating terdiurnal tide (Smith, 2000; Akmaev, 2001; Smith and Ortland, 2001). The pronounced
amplitude peak around DOY 260 observed in SD uniquely stands out, however, possibly owing to the high temporal resolution
offered by the radars. The amplitude peak appears to be an enhancement superimposed on the broad wintertime maximum.
There are a number of mechanisms that can excite a time-localized forcing of TW3, such as non-linear wave-wave interactions
and diurnal tide and gravity wave interactions (Teitelbaum et al., 1989; Miyahara and Forbes, 1991). If conditions are favourable
for any such mechanisms to come into effect around DOY 260 remains to be examined. Here we note that traditional single
point radar measurements of the 8 hour wave are prone to contamination by gravity waves, for which eight hours falls in
the middle of the typical mid- to high-latitude spectrum at MLT heights (e.g., Federico Conte et al., 2018). Gravity wave
contamination is expected to be comparatively low for the TW3 tidal mode retrieved from SD, however, since the horizontal
scale of gravity waves is much smaller than the longitudinal extent covered by the radars.

The DW1 tidal mode shows considerably more short-term and interannual variability, and a different seasonal behavior
between the zonal and meridional component. A possible cause of this is that DW1 has a relatively short vertical wavelength.
Whereas the semidiurnal and terdiurnal tides have a vertical wavelength on the order of 100 km in the MLT (Chapman and
Lindzen, 2012; Yuan et al., 2008; Smith, 2000), the diurnal tide has a vertical wavelength on the order of 25-35 km (e.g.,
Avery et al., 1989). The vertical wavelength of DW1 is therefore much nearer to the vertical average represented by the SD
meteor winds, which can cause DW1 to partly cancel out over the meteor echo range. In addition, DW1 is the tidal mode
most susceptible to contamination by non-migrating tides, given that the 180 degrees of longitude aliases DW1 and the diurnal
oscillation in the mean wind (D0). Although no evidence of such cross-contamination is found in the NAVGEM-HA sampling
experiments, and D0 amplitudes in NAVGEM-HA (NAVGEM-360) never reach above 4.0 ms$^{-1}$ at this latitude. Lastly, DW1
is likely to be the tidal mode that is most strongly affected by the diurnal cycle of meteor echoes (Hussey et al., 2000; Tsutsumi
et al., 2009). Nonetheless, the climatology of the meridional component of DW1 shows close agreement with the seasonal cycle
of the anti-symmetric diurnal (1,1) Hough-mode calculated from TIMED Doppler Interferometer (TIDI) and NAVGEM-HA
meridional winds by Dhadly et al. (2018). It is possible that certain diurnal modes are selectively filtered by SD based on their
respective vertical wavelength and that the (1,1) mode is the dominant remaining mode, even though the amplitude of this
mode broadly peaks around 25°N (Chapman and Lindzen, 2012). Future work could go out to investigating if the climatology
of the zonal component of DW1 in SD can also be associated with the diurnal (1,1) Hough mode.

## 6 Conclusion

This study has leveraged the longitudinal coverage of 10 high-latitude SuperDARN (SD) radars to isolate the DW1, SW2, and
TW3 tidal modes from 16 years of hourly meteor wind measurements of the mid- to high-latitude MLT. Based on sampling





experiments with NAVGEM-HA, it is demonstrated that the SD tidal modes are closely representative of the (global) migrating tides along 60°N. The amplitude and phase structure of SW2 and TW3 show a strongly recurring seasonal cycle, whereas DW1 shows considerably more year-to-year variability. Notable observations are that the climatological late summer amplitude maximum of SW2 in the zonal (meridional) wind is 8.3 (6.4) ms$^{-1}$ greater than the mid-winter maximum, and that TW3 is marked by a secondary amplitude peak around DOY 260 that reaches values of $5.3 \pm 1.7$ ms$^{-1}$ in the zonal wind. In addition, DW1 is found to show a bi-modal circular phase polarization relation, where the zonal component leads the meridional during most of the year and vice versa during summer.

Many open questions remain in terms of how tidal variability is coupled to variability in their forcing mechanisms and propagation conditions. For future work, the time series of validated SD tidal measurements presented in this work can serve as a valuable source of data in studying the long- and short-term trends and variability of the migrating tides in the high-latitude MLT. The method and validation steps outlined in this work will also contribute to similar analyses of SD meteor winds from the continuously expanding global network of radars.

*Author contributions.* WEC, PJE and REH developed the concept, while WEC performed the data analysis and wrote the paper. JPM provided the NAVGEM-HA data and contributed section 2.3. PJE, REH and JPM gave feedback on the conceptual development and draft versions of this work.

*Competing interests.* The authors declare that no competing interests are present.

*Acknowledgements.* The authors acknowledge the use of the SuperDARN meteor wind data product. The SuperDARN project is funded by national scientific funding agencies of Australia, China, Canada, France, Japan, Italy, Norway, South Africa, the United Kingdom, and the United States. Data are available from Virginia Tech at vt.superdarn.org. Development of NAVGEM-HA was supported by the Chief of Naval Research and the Department of Defense High Performance Computing Modernization Project.

The current research was partly funded by the Research Council of Norway/CoE under contract 223525/F50.



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
