# Peer review of "Migrating tide climatologies measured by a high-latitude array of SuperDARN HF-radars"

_Annales Geophysicae, 2020_

## Referee Comment (RC1) · Anonymous Referee #1 · 18 Aug 2020

Review: Migrating tide climatologies measured by a high-latitude array of SuperDARN HF-radars by van Caspel et al.

This work uses an array of 10 high-latitude superDARN HF-radars to analyze migrating tides (DW1, SW2, TW3). SuperDARN radars can cover 180 degrees of longitudes and allow tidal analysis. Their modeling works indicated that missing parts of longitudinal coverage (radar sampling) do not show significant influences of their tidal analysis method. Their method provide higher-frequency tidal variability compared to satellite observations, and it is useful for future tidal studies. I recommend publication after several minor/major revisions.

[Figure]

Comment (1) This is my main comment for this work. Do SuperDARN radar tides compare well with your model (NAVGWM-SD)? I saw NAVGEM-SD and NAVGEM -360 comparison, and results look great. Authors mentioned in the section 2.3, NAVGEM – HA show good agreement with tides and winds from previous radars and satellite observations. I am wondering if your tides show good agreement with NAVGEM. Can you add SuperDARN radar tidal results in Figure 5 along with NAVGEM-SD and NAVGEM-360? Or can you show us comparisons between modeling work and SuperDARN radar observed tides?

Comment (2) It is hard to see where are DOY 250, 260, 365 etc mentioned in the page 4-5 for Figures 2-3. Also it is hard to see where is "late summer" and "mid-winter" from Figure 2 and 3. Would you add vertical lines for every year (currently every two years)? Can you also specify "late summer' and "mid-winter"(which months are you talking about?).

Figures 2,3, and 4: Authors discussed a lot about DOY 260. Would you indicate DOY 260 in some of your figures? X-axis is years and it is hard to see from Figures 2-3.

Comment (3) Figure 5: What are you plotting? Zonal wind? Or meridional wind? (I think it is zonal wind, but it is not clear).

Comments (4) Line 185. Authors discussed that radars can see high-temporal resolutions, resulting in the peak around DOY 260. Would you discuss more about this? What are temporal resolution of previous terdiurnal tide work?

---

## Author Comment (AC1) · 3 Sep 2020

The authors would like to thank the reviewer for their comments on our work. We address each comment individually by stating the original Refereree Comment (RC) followed by the Author's Response (AR).

RC1: This is my main comment for this work. Do SuperDARN radar tides com- pare well with your model (NAVGWM-SD)? I saw NAVGEM-SD and NAVGEM -360 comparison, and results look great. Authors mentioned in the section 2.3, NAVGEM – HA show good agreement with tides and winds from previous radars and satellite ob- servations. I am wondering if your tides show good agreement with NAVGEM. Can you

add SuperDARN radar tidal results in Figure 5 along with NAVGEM-SD and NAVGEM-360? Or can you show us comparisons between modeling work and SuperDARN radar observed tides?

AR1: The purpose of including the NAVGEM-HA analysis was to validate the Super-DARN tidal analysis method. That is, that the combination of radars, each measuring the tidal oscillations at different locations, can be used to extract the unambiguous migrating components without bias due to the discrete spatial sampling of SuperDARN. We did not include a direct comparison between SuperDARN and NAVGEM-HA on the basis that the SuperDARN tides represent a broad vertical average (calculated from a Gaussian meteor echo distribution centered on ∼100 km altitude with a FWHM of 25-35 km), whereas NAVGEM-HA can be used up to an altitude of ∼90 km for tidal analysis. The SuperDARN meteor winds therefore represent tidal measurements of a region that is largely outside of the NAVGEM-HA model domain. A detailed comparison between the modeled and observed tides would require the model domain to be extended up to ∼ 125 km altitude below the sponge layer, such that vertically averaged model winds can be compared to those measured by SuperDARN. Nonetheless, we think the inclusion of the SuperDARN tidal modes in Figure 5, as suggested by the reviewer, is a worthwhile addition, not in the least because it demonstrates more clearly that the modeled and measured tidal modes share similar seasonal characteristics, further justifying the use of NAVGEM-HA to validate the SuperDARN tidal analysis method. The SuperDARN measurements have therefore been included in Figure 5 of the revised manuscript, and a brief description of the above reasoning has been included as a third paragraph in section 4.1.

RC2: It is hard to see where are DOY 250, 260, 365 etc mentioned in the page 4-5 for Figures 2-3. Also it is hard to see where is "late summer" and "mid-winter" from Figure 2 and 3. Would you add vertical lines for every year (currently every two years)? Can you also specify "late summer' and "mid-winter"(which months are you talking about?).

AR2: Throughout the manuscript, text referring to certain time periods has been

changed so as to be more clear about what time period is being referred to in Figures 2-4 (e.g., 'mid-winter' (December - January)). Vertical lines have been added for each year in Figures 2-3, and Figure 4 now shows DOY on its x-axis (see AR3).

RC3: Authors discussed a lot about DOY 260. Would you indicate DOY 260 in some of your figures? X-axis is years and it is hard to see from Figures 2-3.

AR3: The labeling on the x-axis of Figure 4 has been changed to Day Of Year (DOY), and a vertical line at DOY 265 is included in the bottom left panel showing the climatology of the amplitude of the migrating terdiurnal tide (TW3, bottom left panel). What was referred to as the DOY 260 amplitude peak is now referred to as the DOY 265 amplitude peak, to more precisely reflect its exact timing.

RC4: Figure 5: What are you plotting? Zonal wind? Or meridional wind? (I think it is zonal wind, but it is not clear).

AR4: Figure 5 plots the migrating tidal modes in the meridional wind, which is now clarified in the figure caption.

RC5: Authors discussed that radars can see high-temporal reso- lutions, resulting in the peak around DOY 260. Would you discuss more about this? What are temporal resolution of previous terdiurnal tide work?

AR5: To the best of the author's knowledge, previous observational studies capable of unambiguously isolating the migrating component of the terdiurnal tide in the northern hemisphere mid- to high-latitude MLT region have exclusively relied on satellite observations. Such observations are limited to temporal resolutions of monthly timescales. For example, Smith (2000) combines UARS data over two yaw cycles (70 day average) to retrieve TW3 tidal amplitudes at 60N. As a result, features such as the DOY 265 maximum observed by SuperDARN are not distinguishable in their figure 2. Other studies capable of isolating the mid- to high-latitude migrating terdiurnal tide have used SABER/TIMED satellite data (e.g., Moudden et al., 2013; Pancheva et al., 2013), but

in addition to being limited to 20- to 60-day means, they only report temperature tides, which further complicates the comparison with SuperDARN.

Model studies seem to show mixed results in terms of the temporal resolution of their tidal analysis and the effective temporal resolution of their results. The cited works by Akmaev (2001) and Smith et al. (2001) describe model results showing a qualitatively similar seasonal cycle as the observed SuperDARN TW3 (i.e., a broad amplitude maximum in the zonal and meridional winds during winter). These studies have used monthly mean specifications of the background atmosphere. For example, in Figure 2 of Smith et al. (2001), no DOY 265 peak is distinguishable at 97 km altitude. The study by Yue et al. (2013) employs a model that is configured using a mixture of daily mean and monthly mean atmospheric background fields. They report monthly mean TW3 tidal amplitudes at 110 km altitude, where a DOY 265 amplitude peak is not visible at 60N (their Figure 3).

However, the TW3 model study using the Canadian Middle Atmospheric Model by Du et al. (2010), where a monthly-mean sliding window is used to analyze internally generated 3-hourly model winds, does show an amplitude peak around DOY 265 around 100 km altitude at 60N (their figure 3). Because of this qualitative agreement with the SuperDARN observations, reference to their model study has been included in the discussion section of the revised manuscript.

Smith, A. K. (2000). Structure of the terdiurnal tide at 95 km. Geophysical Research Letters, 27(2), 177–180. doi:10.1029/1999gl010843

Akmaev, R. A. (2001). Seasonal variations of the terdiurnal tide in the mesosphere and lower thermosphere: A model study. Geophysical Research Letters, 28(19), 3817–3820. doi:10.1029/2001gl013002

Smith, A. K., & Ortland, D. A. (2001). Modeling and analysis of the structure and generation of the terdiurnal tide. Journal of the atmospheric sciences, 58(21), 3116-3134. doi:10.1175/1520-0469(2001)058<3116:MAAOTS>2.0.CO;2

Moudden, Y., & Forbes, J. M. (2013). A decade-long climatology of terdiurnal tides using TIMED/SABER observations. Journal of Geophysical Research: Space Physics, 118(7), 4534–4550. doi:10.1002/jgra.50273 Yue, J., Xu, J., Chang, L. C., Wu, Q., Liu, H.-L., Lu, X., & Russell, J. (2013). Global structure and seasonal variability of the migrating terdiurnal tide in the mesosphere and lower thermosphere. Journal of Atmospheric and Solar-Terrestrial Physics, 105-106, 191–198. doi:10.1016/j.jastp.2013.10.010

Pancheva, D., Mukhtarov, P., & Smith, A. K. (2013). Climatology of the migrating terdiurnal tide (TW3) in SABER/TIMED temperatures. Journal of Geophysical Research: Space Physics, 118(4), 1755–1767. doi:10.1002/jgra.50207

Du, J., & Ward, W. E. (2010). Terdiurnal tide in the extended Canadian Middle Atmospheric Model (CMAM). Journal of Geophysical Research: Atmospheres, 115(D24). doi:10.1029/2010jd014479

---

## Referee Comment (RC2) · Anonymous Referee #2 · 12 Sep 2020

Recommendation: Publish with minor revisions

This paper presents observations of migrating diurnal (DW1), semidiurnal (SW2) and terdiurnal (TW3) tides in 16 years of high-latitude horizontal winds measured by the SuperDARN network. These are the only direct measurements of mesospheric and lower thermospheric winds from which migrating tides can be defined globally on time scales shorter than satellite 24-hour precession periods (1-2 months). The authors have analyzed the data very carefully, and make a strong case for the fidelity of their tidal retrievals. Aside from a few curious features (e. g., reversal of the polarization of DW1 winds during summer, a TW3 amplitude maximum in October, an SW2 maximum

in September), the tides do not exhibit any particularly interesting behavior. The main strength, and take-home message of the paper is the validity of the analysis. Although straightforward, the robustness of this method was not a foregone conclusion when applied to SuperDARN winds, since the network only spans about 200 degrees longitude. I therefore consider the material worthy of publication, because of the promise of this method for identifying short-term tidal variability, a topic that is highly pertinent to vertical coupling and whole-atmosphere modeling.

This paper is for the most part clearly written and organized. I recommend publication after the authors respond to the following, mostly minor comments.

1. Page 1, line 14: Perhaps replace the term "Hough" (which will be unfamiliar to most readers) with "spherical harmonic"?

2. Page 1, line 23: "yaw cycle intermittency" sounds wordy and opaque. Replace with "slow local time precession".

3. Page 4, lines 76-77: This sentence is incomprehensible. Are you trying to say that "If measurements are not available for both stations at any given time, measurements are excluded in a manner so as to optimize the equidistant longitudinal spread of measurements?"

4. Figures 2-5 need to be enlarged.

5. I suggest showing the climatology first, then the year to year variability.

6. Page 7, line 151: replace "tidal modes" with "tides".

7. Page 8, lines 155-156: Simplify to: "...not lead to significant cross-contamination errors between the migrating tides."

8. Figure 6: Any idea why the RMS difference for SW2 is so much higher than the others?

9. Page 10, line 187: "Should read "Whether conditions are favourable..."

10. Page 10. The nomenclature is confusing. (1,1) is the first symmetric propagating Hough mode. (1,2) is the first antisymmetric propagating mode.

11. The term "mode" refers to the latitudinal structures, or Hough modes. It should not be used to describe the longitudinal wavenumber or frequency. Thus, DW1, SW2, etc. are tides. (1,1) is a mode.

12. Page 10, lines 195- 208. Lots of speculation here about the diurnal winds and how they may be distorted by the SuperDARN "observational filter". Is it feasible to quantify these effects by forward modeling DW1 winds into meteor echoes?

---

## Author Comment (AC2) · 29 Sep 2020

The authors would like to thank the reviewer for their comments on our work. We address each comment individually by stating the original Referee Comment (RC) followed by the Author's Response (AR).

RC1: Page 1, line 14: Perhaps replace the term "Hough" (which will be unfamiliar to most readers) with "spherical harmonic"?

AR1: The authors agree with RC1 in that latitudinal spherical harmonic structures (associated Legendre polynomials) will be more familiar to most readers. However,

since Hough modes form such an integral part of tidal theory (as described in detail in the work of Chapman and Lindzen (2012) cited on line 13), and because Hough modes are also referred to in the discussion, we would still like to make reference to Hough modes in our introduction of the tides. Since both spherical harmonics and Hough modes can be expressed as a combination of either one, as both form a complete set of basis functions, we think the following change to line 14 in the revised manuscript may offer more clarity:

"The tides have a latitudinal spherical harmonic structure (termed Hough modes) and longitudinal zonal wavenumber (S) structure,..."

RC2: Page 1, line 23: "yaw cycle intermittency" sounds wordy and opaque. Replace with "slow local time precession".

AR2: Line 23 has been rewritten to include slow local time precession as a drawback of satellite measurements, while still including yaw cycle maneuvers (e.g., the changing latitudinal coverage of SABER/TIMED every 60 days, and every 36 days for MLS/UARS) as a separate drawback. Line 23 now reads:

"Typical drawbacks associated with satellite measurements arise due to constraints imposed by asynoptic sampling (Salby, 1982), including slow local time precession and yaw cycle maneuvers."

RC3: Page 4, lines 76-77: This sentence is incomprehensible. Are you trying to say that "If measurements are not available for both stations at any given time, measurements are excluded in a manner so as to optimize the equidistant longitudinal spread of measurements?"

AR3: It was our intention to say that, if measurements are available for two closely-spaced (in longitude) stations at any given time, we leave out one of the station's measurements in the fit to Eq. 1, such that the longitudinal spread of measurements becomes more equally spaced over the fitting domain.

For example, if at a certain time there are measurements available from Han, Pyk, Sto, Gbr, Sas and Kod (as shown in Fig. 1), the measurement of Pyk is excluded in the fit to Eq. 1. If we didn't exclude Pyk, the winds measured at the longitude corresponding to the (Sto, Pyk) 'longitude pair' would effectively have a double weight in the least-squares fitting routine, which is undesirable. The measurement of Pyk is therefore excluded in the fit, since the fit is then applied only to measurements that are spaced roughly 50 degrees longitude apart. In the revised manuscript, lines 76-77 have been rewritten to,

"To that end, for the radar pairs closely spaced in longitude, (Ksr, Kod), (Rnk, Kap), and (Sto,Pyk), only one of each of the pairs measurements is used in the fit to Eq. 1, even if data are available for both.".

RC4: Figures 2-5 need to be enlarged.

AR4: Figures 2-5 have been enlarged as well as modified in accordance with the comments of Reviewer 1.

RC5: I suggest showing the climatology first, then the year to year variability.

AR5: We followed the plan of similar climatological papers, where the raw data, with its variability, is presented first. The climatology is then shown to emphasize that only the seasonal variations present in the data have been isolated in the climatology. We then go on to discuss only those features observed in the climatology and draw conclusions from them. If we reverse the order, the flow jumps from climatology, to raw data, to discussing the climatology. While we could reverse the order without prejudicing the results, we would prefer to keep the present order to reflect other climatology papers and to provide a better "readability" factor.

RC6: Page 7, line 151: replace "tidal modes" with "tides".

AR6: "tidal modes" has been replaced with "tides" throughout the text in accordance with RC11.

RC7: Page 8, lines 155-156: Simplify to: "...not lead to significant cross-contamination errors between the migrating tides."

AR7: Lines 155-156 have been simplified in accordance with RC7.

RC8: Figure 6: Any idea why the RMS difference for SW2 is so much higher than the others?

AR8: We are unable to identify any particular reason why SW2 has a RMSE roughly 50% greater than that of DW1 and TW3. For example, no one-to-one relation exists between the yearly mean tidal amplitudes in NAVGEM-360 (9.8, 10.1, and 2.74 ms-1 for DW1, SW2, and TW3, respectively) and the yearly mean RMSE values shown in Figure 6.

RC9: Page 10, line 187: "Should read "Whether conditions are favourable..."

AR9: Line 187 has been updated in accordance with RC9.

RC10: Page 10. The nomenclature is confusing. (1,1) is the first symmetric propagating Hough mode. (1,2) is the first antisymmetric propagating mode.

AR10: For consistency, we now refer to the diurnal (1,1) mode in the same manner as Dhadly et al. 2018, which is the work cited in our discussion on the DW1 tide. The mode is now simply referred to as the Diurnal (1,1) Hough-mode.

RC11: The term "mode" refers to the latitudinal structures, or Hough modes. It should not be used to describe the longitudinal wavenumber or frequency. Thus, DW1, SW2, etc. are tides. (1,1) is a mode.

AR11: "tidal modes" has been replaced with "tides" throughout the text to correctly reflect the distinction between the latitudinal Hough mode structure and longitudinal wavenumber structure of atmospheric tides.

RC12: Page 10, lines 195- 208. Lots of speculation here about the diurnal winds and how they may be distorted by the SuperDARN "observational filter". Is it feasible to

quantify these effects by forward modeling DW1 winds into meteor echoes?

AR12: The various factors that might impact the representation of DW1 in the Super-DARN meteor winds have been further examined in the interim, and the vertical wave-length of DW1 being near to the vertical average represented by SuperDARN winds is likely to be the most impactful. Hence, to avoid unnecessary speculation, the following lines have been removed from the text:

"In addition, DW1 is the tide most susceptible to contamination by non-migrating tides, given that the 180 degrees of longitude aliases DW1 and the diurnal oscillation in the mean wind (D0). Although no evidence of such cross-contamination is found in the NAVGEM-HA sampling experiments, with D0 amplitudes in NAVGEM-360 never reaching above 4.0 ms-1. Lastly, DW1 is likely to be the tide that is most strongly affected by the diurnal cycle of meteor echoes (Hussey et al., 2000;Tsutsumi et al., 2009)."

The quantify the effect of the vertical average "observational filter" of SuperDARN would require the model top of the NAVGEM-HA meteorological analysis system to be extended at least up to ∼125 km altitude. But as discussed in AR1 to the comments of reviewer 1, this is beyond the scope of the current work.